# Physicochemical, Microbiological and Microstructural Characteristics of Sucrose-Free Probiotic-Frozen Yogurt during Storage

**DOI:** 10.3390/foods11081099

**Published:** 2022-04-12

**Authors:** Atallah A. Atallah, Elsayed A. Ismail, Hany M. Yehia, Manal F. Elkhadragy, Abeer S. Aloufi, Dalia G. Gemiel

**Affiliations:** 1Department of Dairy Science, Faculty of Agriculture, Benha University, Moshtohor 13736, Egypt; e.ismail@fagr.bu.edu.eg (E.A.I.); dalia.gamil@fagr.bu.edu.eg (D.G.G.); 2Food Science and Nutrition, College of Food and Agriculture Science, King Saud University, Riyadh 11451, Saudi Arabia; hanyehia@ksu.edu.sa; 3Department of Food Science and Nutrition, Faculty of Home Economics, Helwan University, Helwan 11611, Egypt; 4Department of Biology, College of Science, Princess Nourah bint Abdulrahman University, Riyadh 11671, Saudi Arabia; mfelkhadragy@pnu.edu.sa (M.F.E.); asaloufi@pnu.edu.sa (A.S.A.)

**Keywords:** probiotic-frozen yogurt, sweeteners, bulking agents, quality characteristics, carbohydrate profile, microstructure using scanning electron microscopy

## Abstract

Frozen yogurt is known as ice cream with some properties of yogurt. Frozen yogurts are a rich source of sucrose levels between 15% and 28% of total ingredients. Consumers suffering from lactose intolerance and metabolic syndrome are looking for sugar-free products. The current study investigates the sugar replacements by using sweeteners (stevia, sucralose and sorbitol) on physicochemical, microbiological, microstructural and sensory characteristics of probiotic-frozen yogurt. Four different treatments of probiotic-frozen yogurts were studied (control probiotic-frozen yogurt with sucrose (F1), probiotic-frozen yogurt with stevia (F2), probiotic-frozen yogurt with sucralose (F3) and probiotic-frozen yogurt with sorbitol (F4)). The chemical properties were not significantly present *p* > 0.05) during storage in all treatments. In the F1 treatment, sucrose value was higher (14.87%) and not detected in the F2, F3 and F4 treatments. The highest values of overrun, hardness and viscosity (*p* < 0.05) were detected in the F2, F3 and F3 samples, but the lowest value was detected in the F1 treatment. Total *Str. thermophilus* and *Lb. delbrueckii* ssp. *bulgaricus* counts were gradually decreased (*p* < 0.05) during storage periods. At 1 day, the Bifidobacteria counts ranged from 7.56 to 7.60 log_10_ CFU g^−1^ in all groups and gradually decreased during storage, but these bacterial counts remained viable (>6.00 log_10_ CFU g^−1^) during storage periods up to 60 d. During storage periods, the highest scores of total acceptability were detected in the F3, F4 and F2 treatments. Scanning electron microscopy (SEM) micrographs of all probiotic-frozen yogurt treatments illustrated that the microstructures showed a difference with a fine network, size pores and structure between the frozen yogurt with sweeteners (F2, F3 and F3) and control frozen yogurt (F1).

## 1. Introduction

Frozen yogurt is known as ice cream with some properties of yogurt [1,2]. It outperforms classic frozen dairy products because it is partially hydrolyzed sugar milk content and live from lactic acid bacteria (LAB). Lopez et al. [1] observed that LAB remains stable throughout the shelf life in frozen yogurt. Probiotic frozen yogurt is an ice cream product and it contains many functional and nutritious characteristics. Frozen yogurts are a rich source of protein, sucrose and fat contents [2,3]. The more notable function of sucrose in food and dairy products is its sweet taste, but it also adds to the flavor and consistency. It contributes to the Maillard reaction; it also influences many food and dairy qualities. On the other hand, it should not account for more than 10% of the daily caloric intake [4]. A reformulation of the product by total or partial sugar or sucrose replacement is a more often applied strategy in a larger part of the studied food and dairy categories (chocolate, ice-cream and dessert). However, a gradual replacement of sucrose level could be an interesting approach in some specific cases. The innovative strategy of reducing sucrose level in food and dairy is also reported, and it involves the use of multisensory integration principles [4]. Many sucrose replacement strategies have been reported to reformulate dairy products targeted at children: total or/and partial reduction in sucrose, cross-modal interactions, gradual sucrose replacement and other functional product strategies [4]. However, total or partial reduction in sucrose has been the default strategy to replace the sucrose level in dairy products [4]. The demand for frozen dessert products has currently increased [5], especially for frozen dessert products without sucrose [5]. Sucrose level in frozen dessert products was changed between 9% and 28% of total ingredients [5]. Consumers suffering from lactose intolerance, metabolic syndrome, diabetes and allergies are looking for low-sugar products. Using sweeteners instead of sucrose to produce ice cream and frozen yogurt products fulfills the needs of consumers who pay attention to nutrition and naturally balanced dairy products [6]. The replacement of sucrose with sweeteners to prepare sucrose-free frozen dairy products leads to a loss in freezing point, so there is a need for adjustment of solids. A prior can be compensated by maltodextrin and polydextrose, and later point of freezing depressants (i.e., sorbitol and stevia) [7]. The successful preparation of high-quality nutritional and functional frozen dairy depends on imitation and simulation of the organoleptic attributes of standard dairy products [8]. Carbohydrate-based filler agents have recently been utilized for sugar-free manufacturing due to their negative influence on ice cream on the price, production and shelf life [8].

Sucrose contributes a taste and sweetness to frozen desserts [9]. To prepare low-sugar frozen dairy products and to provide a refreshing ice-cream yogurt for diabetics, it is necessary to replace the sucrose. Sucrose is generally used at the rate of about 15–20%, but the intense sweetener is used at levels of milligrams or ppm only [9]. Hence, the recompense of the dry matter to overcome weak consistency (body and texture) is necessarily needed. The addition of filler agents improves smoothness, creaminess, progresses body and texture and prolongs a mouth feeling acceptable to customers [9].

Sweeteners could be artificial or natural products. Stevia powders (*Stevia rebaudiana*) are characterized as natural ingredients and are about 250–300 times sweeter than sucrose [10]. It is a very low-calorie ingredient (about zero energy), which makes stevia a good substitute for sucrose for diabetes-distressed patients [11]. Stevia products are known as safe enrichment by JECFA, WHO and FDA with comparatively high levels [12,13,14]. Sucraloses are an artificial sweetener, and sucralose sweetness is higher than sucrose (nearly 600 times). They are very low energy (1 g contributes 3.36 calories) [15]. Sorbitol, including sugar-free products, is about 60% sweeter than sucrose (1 g contributes 2.60 calories). Sucralose is considered safe, and it is not digested in human stomach [16].

Probiotic-frozen yogurt is known as frozen dairy, which includes functional and nutritious influences. The sucrose percentages in probiotic-frozen yogurt are changed between 15% and 28% of dry matter. The major aim of the current research was to evaluate a produced sugar-free probiotic-frozen yogurt using sweeteners (stevia, sucralose and sorbitol) and bulking agents (maltodextrin and polydextrose) as replacements for sucrose and to study the physicochemical, microbiological, microstructure and sensory properties of the produced probiotic-frozen yogurt during storage periods.

## 2. Materials and Methods

### 2.1. Materials

Buffaloes’ milk (6.2% fat) was obtained from the herd of Agriculture Faculty, Moshtohor, Benha University, Qalubia, Egypt. Fresh buffaloes’ cream (37.0% fat) was produced in Dairy Science Department, Agriculture Faculty, Moshtohor, Benha University, Qalubia, Egypt. Whole bovine milk powders (26.0% fat, 24.0% protein, 2.5% moisture and 36.0% lactose) extra grade-spray dried produced by (California Dairies, Inc., Fresno, CA, USA). The stabilizer (Sodium carboxy methyl cellulose (CMC)) produced by (El-Nasr Company for Chemicals, Cairo, Egypt). The vanilla produced by (Tag El-Melouk Company for Food Industries, 6th October City, Giza, Egypt). The commercial sugar cane manufactured by (Egyptian Sugar and Integrated Industries Company, Hawmdia, Giza, Egypt). Freeze-dried yogurt starter (FD-DVS, YO-MIX 572) including *Streptococcus* (Str.) *thermophilus* and *Lactobacillus* (Lb.) *delbrueckii* ssp. *bulgaricus* (1:1) was obtained from (Christian Hansen’s Laboratories, Danisco-DuPont Group, Copenhagen, Denmark). *Bifidobacterium* (Bifi.) *bifidium* was obtained from National Research Center, Giza, Egypt. Sorbitol (C_6_H_14_O_6_) produced by (Techno pharmchem, Bahdurarh Company, Bahdurarh, India). The stevia (*Stevia rebaudiana*) powder was produced by (Techno pharmchem, Bahdurarh Company, Bahdurarh, India). The sucraloses were produced by (Tale and Lyle specialist sweeteners, Basingstoke, UK). The maltodextrins were produced by (Heilongjiano Haotian Corn Development Co. LTD, Shihua, China). The polydextroses were provided by (Dalya Foreign Trade Co. LDT, Istanbul, Turkey). Sodium hydroxide (NaOH) and phenolphthalein were obtained from (Oxoid, Basingstoke, UK).

### 2.2. Activation of Yogurt Starter and Bifidobacteria

Freeze-dried yogurt starter and bifidobacteria were prepared in 200 mL of reconstituted skimmed milk (10% *w*/*w*, Oxoid, Basingstoke, UK), and it was incubated at 37 °C for 24 h. Lastly, these cultures (yogurt starter and bifidobacteria, 1:1) were reactivated into 300 mL of buffaloes’ milk and incubated at 37 °C until the pH became 4.6 and refrigerated at 4 ± 1 °C.

### 2.3. Manufacture of Probiotic-Frozen Yogurt

The frozen yogurts were formulated according to Arslaner et al. [17,18]. The frozen yogurt mixture (control) was prepared to include fat (8%), solid not fat (12%), sugar (15%) and stabilizer (CMC, 0.25%). Sugar was replaced by different sweeteners and the decrease in total solids of the frozen yogurt mixture was compensated by the addition of maltodextrin and polydextrose (1:1) with the same percentage of sucrose sugar. The incorporation steps of different probiotic-frozen yogurt mixes produced were shown in Figure 1.

Initial trials were conducted in the frozen yogurt produced with different levels of stevia, sorbitol and sucralose. The results detected that the best value of Stevia was 0.06%, sucralose was 0.03%, and sorbitol was 3% in the frozen yogurt produced by conducting some analyses, such as sensory and texture analyses. Therefore, initial trials were conducted in the produced frozen yogurt with different levels of maltodextrin and polydextrose as filler. Data showed that the best concentration of maltodextrin was 7.5%, and polydextrose was 7.5% in the frozen yogurt manufactured.

The data provided in the current study show means contents of three replicate batches, and all parameters (analyses) were determined in triplicate. The mixes of different frozen yogurt treatments were calculated in Table 1.

### 2.4. Chemical Analyses

#### 2.4.1. Total Solids

Total solids (TS) were estimated using 2–3 g and dried at 105 °C [19]. Total protein (TP) was estimated using the Kjeldahl technique as described by IDF [20] as follows:(1)TP=TN ×6.38
where TP is total protein and TN is total nitrogen.

Fat value was determined by ISO [21] using the method of Gerber. Ash was measured by drying the frozen yogurt (100–105 °C), and dried samples were combusted at 550 °C [22]. Total acidity was calculated by sample titration using sodium hydroxyl (NaOH, 0.1 N) and indicator (Phenolphthalein, 0.1%; Oxoid, Basingstoke, UK) by the method mentioned in the ISO [23].

#### 2.4.2. Sugar Analysis

The carbohydrate analyses were determined by the method mentioned in Arslaner et al. [17]. An amount of 5 g of homogenized frozen yogurt was diluted (20 mL of a water–methanol mix (75:25, *v*/*v*)) and mixed under cooling at 5 °C for 10 min to 5000 g. The supernatants were filtered through filter paper (Whatman No. 1) and filter (0.45 µm, GE Healthcare Life Sciences, New York, NY, USA), in the same order. The prepared filtrates were put in vials (2 mL) and stored at −20 °C. The filtrates were determined using high-performance liquid chromatography technique ((HPLC), LC-10A Series; Shimadzu, Tokyo, Japan). The analyses were estimated with HPLC equipment, refractive-index detector (RID-10A) and acetonitrile–water mix (80:20, *v*/*v*; 2 mL min^−1^ flow rate). The injection volumes were 20 µL, and the temperature of column was adjusted to 40 °C. The carbohydrates (lactose, fructose, glucose, galactose and sucrose) were calculated by comparing their sugar standards by times of retention.

### 2.5. Microbiological Chacracteristics

The enumeration of *Str. thermophilus* and *Lb. bulgaricus* was determined by the method mentioned in the procedures outlined by the ISO [24]; by pour plating on M17 agar (Merck, Darmstadt, Germany) and MRS agar (Merck, Darmstadt, Germany), respectively, and incubated for 72 h at 37 °C in anaerobic jar AnaeroGen (Oxoid, Basingstoke, UK) of *Lb. bulgaricus* and aerobic condition of *Str. thermophilus*. Bifidobacteria counts were enumerated in RCA (pH 7.1) plus 1 µL mL^−1^ dicloxacillin (Oxoid, Basingstoke, UK) after incubation for 72 h at 37 °C in anaerobic jar [25]. The coliforms were determined as described by APHA [26]. Yeasts and molds counts were performed as described by IDF [27]. The Total aerobic mesophilic bacteria (TAMB) count was carried out in plate count agar (Oxoid) incubated at 30ºC for 48 hours; at 5ºC for 10 days for total psychrotrophic bacteria count. All bacterial counts were enumerated in duplicate. The numbers were tableted as logarithm colony forming units per gram (log_10_ CFU g^−1^).

### 2.6. Physical Analyses

The WPG (Kg) of frozen yogurt mixes, before whipping and freezing and frozen yogurt treatments after whipping and freezing were determined according to Marshall and Arbuckle [28], by multiplying the specific gravity (SG) by the factor of 3.786 as follows:(2)WPG of mix before freezing (Kg)=SG of mix before freezing × 3.786
(3)WPG of frozen yogurt after freezing (Kg)= SG of frozen yogurt after freezing × 3.786

The freezing point of frozen yogurt treatments after whipping and freezing was measured as described by Marshall et al. [29].

Viscosity was estimated in frozen yogurt mixes before whipping and freezing by the method mentioned in the Brookfield measurement, and it was performed at 5 °C with a spindle (50 rpm, No. #07), and the measured values were recorded after 30 s of rotation to ensure a steady reading in 250 mL cup, viscosity calculated as centi-Poise (cP). The overrun of frozen yogurt samples was determined using Akin et al. [30] as follows:(4)Overrun (%)=[(A−B)/B]×100
where A is weight of mix volume before whipping and freezing, and B is weight of the same volume of frozen yogurt sample after whipping and freezing.

The weight of melted frozen yogurt treatments after whipping and freezing expressed as % of initial weight of frozen yogurt was measured as described by Arndt and Wehling [31]. An amount of 100 g of frozen yogurt sample was put into wire mesh (6 pores /cm^2^). Melted frozen yogurt was determined every 15 min.

The hardness of frozen yogurt after whipping and freezing was measured using a Universal Testing Machine (TMS-Pro, Tokyo, Japan) with 250 lbf load cell and linked to a computer program Texture Pro^TM^ (Texture Pro^TM^, program, DEVTPA with the hold). A flat rod probe (49.95 mm) was used to uniaxially compress the samples (50 g) to 50% of their original height. The probe was proved to a speed 60 mm s^−1^; trigger force 1 N, deformation 25%, temperature at −20 °C and holding time 2 s between cycles. The hardness value by Universal Testing Machine was expressed as Newton (N).

### 2.7. Sensory Characteristics

The sensory characteristics of probiotic-frozen yogurt were applied by the method of Khalil and Blassy [32]. A panelist composed of 15 members of the Department of Dairy Science, Benha University, Egypt, was assembled. Fifty grams of frozen yogurt were submitted to the 15 panelists. The panelists determined the sensory characteristics (color and appearance (10 points hedonic scale), structure and consistency (10 points hedonic scale), taste and odor (10 points hedonic scale), icy structure (10 points hedonic scale), melt in mouth (10 points hedonic scale), gummy structure (10 points hedonic scale), total acceptability (10 points hedonic scale)) in all treatments. The probiotic-frozen yogurts were numbered and coded.

### 2.8. Microstructure Using Scanning Electron Microscopy

Scanning electron microscopy (SEM) of probiotic-frozen yogurt was determined in National Research Centre, Giza, Egypt. The sample was incorporated as reported by Jaya [33]. The sample was put on the iron core and covered with a thin layer of gold in a vacuum chamber for 40 s. The samples were dried in an air-tight desiccator, including silica gel. At least nine images were read by SEM (FEI Company, Eindhoven, The Netherlands) model quanta 250 FEG (field emission gun) attached with EDX unit (energy dispersive X-ray analyses). The images were measured at an excitation voltage of 20 kV at different magnifications ranging between 500 and 6000 and working distance varying between 13.7 and 14.2 mm.

### 2.9. Statistical Analysis

Data of this study were analyzed by ANOVA. Differences between factors were recorded significant at *p* < 0.05. In this experiment, we included two following factors: the first is the treatment (F1, F2, F3 and F4) and the second is the storage period (1, 20, 40 and 60 days). The results were tabled as the Means ± SE. Quantitative PCR (qPCR) datasets were measured using SAS PROC GLM (version 6.12, SAS Institute Inc. Cary, NC, USA) [34]. The static model in this study is as follows:(5)$$Y_{\mathrm{ij}}\;=\;µ\;+\;T_{i}+\;e_{\mathrm{ij}}$$
where Y_ij_ is the variable of dependent, µ is overall mean, T_i_ is treatment effect (i = 1, …, 7), e_ij_ is standard residual error.

## 3. Results

### 3.1. Chemical Properties

The chemical characteristics of probiotic-frozen yogurt are presented in Table 2. Chemical properties did not show significance in (*p* > 0.05) all treatments and during storage periods. The addition of sweeteners and filler agents was not significantly recorded in the protein, fat, total solids, ash, titratable acidity and total carbohydrate contents of probiotic-frozen yogurt treatments during storage periods (*p* > 0.05). The total solids changed between 35.18% and 35.26% (*w*/*w*) during 60 d in all treatments. Protein levels varied from 4.13% to 4.19% (*w*/*w*) in all samples during storage. Fat content ranged from 8.12% to 8.16% (*w*/*w*) in all treatments during storage. Ash values changed between 1.07% and 1.11% (*w*/*w*) in all groups during storage. Titratable acidity level ranged from 0.44% to 0.45% (*w*/*w*) in all groups. The replacement of sucrose with filler agents and sweeteners did not affect the chemical properties of frozen yogurts.

### 3.2. Carbohydrate Profiles

Table 3 presents the carbohydrate profile values for all probiotic-frozen yogurt treatments. Sucrose replacement with stevia, sucralose and sorbitol powders had significantly (*p* < 0.05) influenced fructose and sucrose levels in the samples. Glucose values ranged from 0.28% to 0.84% in all treatments. In control samples, fructose value was 0.40% and not detected in probiotic-frozen yogurts with stevia, sucralose and sorbitol. The highest sucrose level was found in the control treatment (F1) compared to probiotic-frozen yogurts with stevia, sucralose and sorbitol. Sucrose value was 14.87% in control samples and not detected in probiotic-frozen yogurts with stevia, sucralose and sorbitol. The addition of stevia, sucralose and sorbitol had not influenced glucose, galactose and lactose ratios in probiotic-frozen yogurt groups (*p* > 0.05). The level of galactose changed between 0.18% and 0.21% in all groups. The lactose contents ranged from 3.95% to 4.01% in all samples.

### 3.3. Physical Properties

#### 3.3.1. Weight per Gallon (WPG)

The WPG levels of frozen yogurt ranged from 2.73 to 2.76 kg in all treatments. The WPG values were not significantly (*p* > 0.05) represented in all groups (Table 4). The replacement of sucrose with stevia, sucralose, sorbitol and bulking agents did not affect the SG and WPG of the frozen yogurt treatments.

#### 3.3.2. Viscosity

The viscosity level of frozen yogurt mixtures before whipping and freezing is influenced by some factors, including the stabilizer, protein, fat, filler agent and the ingredient quality added. Table 4 presents the viscosity values of all frozen yogurt mixes before whipping and freezing. All frozen yogurt groups before whipping and freezing showed significance (*p* < 0.05) in viscosity values. The viscosity ranged from 90.95 to 95.68 cP in all mixes. The lowest values of viscosity were shown in the control samples (F1), while the highest values of viscosity were detected in the F2, F3 and F4 samples.

#### 3.3.3. Overrun

The overrun values of frozen yogurt are recorded in Table 4, where the overrun was significantly recorded (*p* < 0.05) in all treatments. The overrun values ranged between 57.10% and 59.90% in all frozen yogurts. The overrun level of control sample was lower than the other treatments. The higher increase in overrun value of the F2, F3 and F4 treatments increased with the higher prolongation level of viscosity.

#### 3.3.4. Freezing Point

The freezing point of frozen yogurt treatments is recorded in Table 4. The freezing point of frozen yogurt varied from −2.65 °C to −2.75 °C for all treatments. All frozen yogurt samples were not significantly detected (*p* > 0.05) at freezing point. The freezing point was not influenced by sucrose replacement with sweeteners.

#### 3.3.5. Hardness

The hardness values ranged from 50.32 N to 52.54 N in frozen yogurt treatments (Table 4). The hardness values were significantly recorded in all frozen yogurt samples (*p* < 0.05). The lowest value of hardness was measured in F1. The highest value of hardness was determined in the F2, F3 and F4 treatments.

### 3.4. Melting Value

Figure 1 represents the melting value for frozen yogurt up to 60 d at −20 °C. The melting % was significantly recorded (*p* < 0.05) between all frozen yogurt samples during storage periods. At 1 d, the melting value varied from 11.60 g 100 g^−1^ to 89.00 g 100 g^−1^ in all frozen yogurts. At 60 d, the melting value varied from 10.20 g 100 g^−1^ to 98.80 g 100 g^−1^ in all samples. An increase in the values was detected in F1 treatment followed by a decrease in the F2, F3 and F4 treatments.

### 3.5. Microbiological Properties

Total aerobic mesophilic bacteria (TAMB), psychrotrophic bacteria, *Str. thermophilus*, *Lb. bulgaricus* and bifidobacteria counts are shown in Table 5. The count of frozen yogurt samples at 1 d ranged from 5.10 to 5.20 log_10_ CFU g^−1^ for TAMB in all groups. These counts were lower during prolonged storage periods (*p* < 0.05), and the minimum counts were recorded after 60 d in all frozen yogurts. The number of psychrotrophic bacteria of frozen yogurt samples at 1 d ranged from 2.18 to 2.26 log_10_ CFU g^−1^ in all groups. These counts were lower during prolonged storage periods (*p* < 0.05), and the minimum counts were observed after 60 d in all treatments. The count of total *Str. thermophilus* and *Lb. bulgaricus* were lower during prolonged storage periods (*p* < 0.05) and reached their minimum counts after 60 d in all treatments.

Bifidobacteria (*Bifi*. *bifidium*) count at 1 d changed between 7.56 and 7.60 log_10_ CFU g^−1^ in frozen yogurt samples. The counts were lower during prolonged storage periods (*p* < 0.05) and reached their minimum counts after 60 d in all treatments. At 60 d, the count varied from 6.00 to 6.08 log_10_ CFU g^−1^ of all groups, but the Bifidobacteria counts remained viable (>6.00 log_10_ CFU g^−1^) at 60 d.

Coliform, yeasts and molds numbers were not observed in all frozen yogurt treatments during the storage periods.

### 3.6. Sensory Characteristics

The organoleptic evaluation of the frozen yogurt is shown in Figure 2. The color and appearance scores showed significance (*p* < 0.05) in all frozen yogurt treatments during storage. The structure and consistency values were significantly recorded (*p* < 0.05) at 1 d and during the storage periods. The taste and odor scores were improved during storage (*p* < 0.05). The melt-in-mouth scores were significantly shown in all frozen yogurts during storage (*p* < 0.05). The total acceptability was improved (*p* < 0.05) during storage. At 1 d of storage, the highest score of total acceptability was detected in the F3, F4 and F1 samples compared with the F2 samples. After 20 d, the highest score of total acceptability was detected in the samples F3 and F4 compared with the other treatments. Total acceptability scores showed overall quality during storage (*p* < 0.05). After 60 d, the highest score of total acceptability was detected in the F3 and F4 treatments compared with the other treatments. In general, total acceptability recorded the highest score in F3 and F4, but the lowest value was recorded in F2 and F1 during storage periods.

### 3.7. Microstructure

SEM micrographs of frozen yogurt samples are shown in Figure 3. The SEM shows the addition effects of stevia, sucralose, sorbitol and bulking agents on frozen yogurt microstructure. In frozen yogurts with stevia, sucralose, sorbitol and bulking agents, the gel appeared to change its structure with a fine filament, and it included very small pore sizes. Additionally, the frozen yogurt containing stevia, sucralose, sorbitol and bulking agents appeared to be a firm and bound gel, and the gel appeared to be regular long casein filaments. On the contrary, the control samples showed an incoherent and unconnected gel and matrix containing large pores and spaces. The gel of control samples appeared irregular in shape, with short casein filaments and individualized casein filaments. The microstructure of control samples showed heterogeneity in the size of pores. The addition of sweeteners and filler agents gave a fine filament and microstructure, providing a number of very low-size pores.

## 4. Discussion

Probiotic-frozen yogurts are a frozen dairy that includes functional and nutritious influences. Sucrose values in probiotic-frozen yogurt range between 15% and 30% of the total ingredients. The replacement of sucrose with sweeteners in the manufacture of frozen dairy desserts can address the issues of customers who focus on the functional and normal adjusted dairy sources [35]. The probiotic-frozen yogurts are mainly formulated using sweeteners and filler agents. The addition of sweeteners to frozen dairy products is very helpful, as it provides sweetness without sugar helps in weight reduction and adjusts diet.

Chemical parameters did not appear significantly among the probiotic-frozen yogurts during storage; this resulted in ratio stability of dry matters, protein and lipid in the samples. In similar research, the compositional properties were not significantly observed in the frozen dairy desserts [35] when sugar limit was lowered by reduction in sugar using the maltodextrin as filler agent and sorbitol. Abdou et al. [36] observed no variances in the chemical properties of ice cream samples when sucrose was lowered in a mix of aspartame and polydextrose. Pinto and Dharaiya [37] noted no significant influence on the TS, fat, protein and titratable acidity values in the frozen dessert samples.

The highest content of sucrose was found in the control compared with the probiotic-frozen yogurt with stevia, sucralose and sorbitol. In control samples, the sucrose value was 14.87% and not detected in probiotic-frozen yogurt with stevia, sucralose and sorbitol. This may be the whole alternative to sugar with sweeteners. In general, sugar has several defects due to its big glycemic index (GI) that facilitates the progression of several metabolic diseases [35,38]. Yogurt ice cream replaced with stevia as sweetener can be a replacement product for diabetes mellitus sufferers. Likewise, several studies have observed that the use of stevia as a sugar substitute did not significantly affect acidity values during storage in frozen yogurt sweetened with honey, stevia and sugar [17]. The lactose levels varied from 3.95% to 4.01% in all probiotic-frozen yogurt treatments.

Sugar alternatives, namely stevia, sucralose, sorbitol, did not influence the WPG values of the frozen yogurt treatments. There were no significant differences among all the probiotic-frozen yogurt treatments, which may be due to the ratio stability of dry matter, protein and lipid in the mixtures.

The viscosity values increased in probiotic-frozen yogurt with stevia, sucralose and sorbitol (F2, F3 and F4) compared with the control samples (F1). The addition of sweeteners and filler agents increased the viscosity value in the frozen yogurt mixes. This is because the addition of polydextrose and maltodextrin helps bind the water and form a heavy gel, which leads to a high viscosity in low-sucrose frozen products [39]. Maltodextrin can play a key role in the beneficial viscosity of the mixes [40]. These results confirm those of Pinto and Dharaiya [37] who noticed that the viscosity increased in fat and sugar alternatives, namely sweeteners and filler agents, of frozen desserts compared with the control.

The highest value of overrun was recorded in the F2, F3 and F4 treatments, and these treatments recorded the highest value of viscosity. Hence, the sweeteners alternative to sugar (F2, F3 and F4) had improved overrun values of frozen yogurts. Verma [35] detected an increase in viscosity values with high maltodextrin level as filler agent in frozen dairy. These data show the same trend as those of Pinto and Dharaiya [37], who reported that the highest content of overrun was recorded in low-fat sugar-free frozen products. The freezing point was not influenced by sweeteners alternative to sucrose. Whelan et al. [41] noticed that the freezing point was not influenced by commercial sweeteners alternative to sucrose and polydextrose of low GI ice creams.

The highest value of hardness was observed in the F2, F3 and F4 treatments compared with the control samples (F1). This increase may be due to the high values of viscosity in frozen yogurt. Tharp [39] reported that the hardness value was dependent on the consistency and viscosity of the frozen dessert. The addition of filler agents increased the hardness value of the frozen yogurt products compared with the control [39]. These data are confirmed by Verma [35], who observed an increase in hardness value with increased addition of maltodextrin as bulking agent in frozen desserts using sweeteners. The hardness is influenced by some factors in ice cream: overrun, size of ice crystalline, destabilization of fat, ice stage quantity and texture properties of the ice cream mixtures [42].

An increase in the content of melting was recorded in the control (F1). A decrease in the melting value was observed in F2, F3 and F4. This may be due to the increase in viscosity and freezing point. The decrease in melting value may be due to the ability of the polydextrose and maltodextrin to bind free water and increase the viscosity of the frozen yogurt mixtures. Besides that, many studies have shown that sucrose with minimizer molecular have a lower melting % as compared to those with bigger molecular [42]. Pinto and Dharaiya [37] noticed that the melting rate recorded an increase in melting of the control followed by a decrease in frozen dessert with sorbitol and bulk agents (polydextrose and maltodextrin).

Total *Str. thermophilus* and *Lb. delbrueckii* ssp. *bulgaricus* counts decreased gradually and reached their minimum counts after 60 d in all treatments. These data are confirmed by Arslaner et al. [17], who reported that the counts of *Str. thermophilus* and *Lb. delbrueckii* ssp. *bulgaricus* decreased gradually during storage in the frozen yogurt. *Str. thermophilus* and *Lb. delbrueckii* ssp. *bulgaricus* counts decreased gradually in probiotic-frozen yogurt during storage [43].

The count of total Bifidobacteria (*Bifi. bifidium*) decreased gradually in all samples during storage, but the counts of Bifidobacteria remained viable (> 6.00 log_10_ CFU g^−1^) up to 60 d. The count of *Bifidobacterium lactis* in frozen products was 10.72 logs CFU g^−1^ before freezing, and the viability decreased during storage [43]. The LAB counts were slightly decreased during storage periods for the frozen yogurt [2]. Probiotic bacteria require being active and viable to a level of at least 6.00–7.00 log_10_ CFU g^−1^ in the fresh dairy products and after storage [44].

Yeasts and molds counts were not detected in all frozen yogurt during storage periods. These data are confirmed by Arslaner et al. [17], who noticed that yeasts and molds were not detected in ice-cream yogurt with honey, stevia and sucrose. Coliforms, yeasts and molds counts in the ice cream were not detected [16].

The reduction in LAB viability during the step of freezing may be due to stress resulting from the mixing of liquid and dry ingredients and development of organic acids during time of incubation [45]. In the freezing step, the probiotic-frozen yogurt mixtures are persistently scratched against the cylindrical surface using freezer blades [45]. The freezing step not only mixes the air properly but also breaks big air bubbles into lower-size ones [45]. This action shock may interfere with integrity of the viability probiotic causing cells to decline [45]. Frozen products are fluffy products, composed of 50% to 100% air. This probiotic is mostly facultative anaerobic and/or micro-aerophilic. The various osmotic freeze shocks and heat shocks in the production of ice cream are lethal to the microbes [46].

The taste and odor, structure and consistency and total acceptability was improved during storage. At 1 d of storage, the highest numbers of structure and consistency, taste and odor and total acceptability were detected in the F3 and F4 samples. At 60 d, the highest numbers of structure and consistency, taste and odor and total acceptability were detected in the F3, F4 and F2 treatments. Verma [35] reported an increasing trend in the organoleptic numbers with high maltodextrin amount with respect to the structure and consistency, melting in mouth, flavor and total acceptability. Additionally, sensory numbers were higher in stevia alternative to sugar in the ice cream [47].

The changes in microstructure between F2, F3 and F4 samples and control (F1) could have increased due to changes in particle characteristics. The F2, F3 and F4 samples appeared as firm and bound gel. Additionally, the gel appeared to be in regular long casein filaments. On the contrary, the control (F1) samples showed an incoherent and unconnected gel and matrix containing large pores and spaces. The gel of control samples appeared irregular in shape, with short casein filaments and individualized casein filaments. The addition of bulk agents resulted in smoothness, creaminess, improved texture and microstructure [48]. Agustini et al. [49] noticed that ice cream with spirulina resulted in changes to the body with a fine gel, providing very low pores. These results confirm those of Atallah et al. [50], who observed that sweeteners alternative to sucrose in the ice cream resulted in changes to the structure, with a fine gel providing very low pores compared with the control treatment.

## 5. Conclusions

In this study, four different probiotic-frozen yogurts were produced (F1, F2, F3 and F4). The physicochemical, microbiological, microstructural and sensory characteristics of frozen yogurt are important quality characteristics. The chemical characteristics were not significantly affected by sweeteners alternative to sugar compared with the control samples. The values of overrun, viscosity and hardness were significantly present in all the treatments. The *Str. thermophilus* and *Lb. bulgaricus* counts decreased gradually with prolonged storage periods (*p* < 0.05). In all treatments, *Bifidobacterium bifidium* counts varied from 7.56 to 7.60 log_10_ CFU g^−1^ at 1 d, and these counts were gradually lower with prolonged storage periods (*p* < 0.05), but the counts remained viable (>6.00 log_10_ CFU g^−1^) up to 60 d. In the F1, F2 and F3 treatments, the highest values of total acceptability were detected during storage periods. SEM micrographs in all frozen yogurt treatments exhibited differences in the microstructure. The microstructure characteristics were improved in probiotic-frozen yogurts with stevia, sucralose and sorbitol compared with the control sample. However, further study is necessary to further replace the fat and sugar to produce the low-calorie frozen yogurt and determine the maintenance quality of these products. Nowadays, Egyptian customers are more aware of their health status and, hence, are conscious of fat- and sugar-free frozen yogurt diets.

## Data Availability

Data available upon request.

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
