# Peer review of "Physicochemical, Microbiological and Microstructural Characteristics of Sucrose-Free Probiotic-Frozen Yogurt during Storage"

_foods, 2022, doi:10.3390/foods11081099_

Round 1
Reviewer 1 Report
Dear Authors,
The paper penned by Atallah et al. has scientific potential, but in the current form it can not be accepted. The text has to undergo rigorous English editing by native speaker in terms of grammar, vocabulary and punctuation as it is not a reader friendly. There are a lot of errors in the text, there is no point in listing them all, I will just to name a few. There are many commas in the text that are used unjustifiably. Also, please don't use this many semicolons, combine sentences with each other more neatly. I recommend major revision.
Some other comments to improve the manuscript:
Title - you may cross out sensory as these properties fall into physicochemical category
Please do not repeat the same sentences from the summary in the text, some of the same content is duplicated several times.
introduction
Lines 42-46 - Probiotic frozen yogurt products are considered to be a frozen dairy product that includes healthy and nutritious characteristics of ice cream mixed with probiotic-yogurt. Frozen yogurt is a rich source of protein, sugar, and fat levels. The demand of frozen desserts products has currently increased especially frozen desserts products without sucrose are increasing [4] .
– please reformulate these sentences, they are very word heavy
Generally, there are many such sentences throughout the text, and in these context the article as a whole needs comprehensive improvement. in addition, the introduction is written chaotically, often there is no logical sequence between the sentences
[4] - reference no 4, concerns vegan ice cream, what does vegan ice cream have to do with the content quoted above? Please cite articles that really do, for example:
Di Monaco, R., Miele, N. A., Cabisidan, E. K., & Cavella, S. (2018). Strategies to reduce
sugars in food. Current Opinion in Food Science, 19, 92-97.
Nastaj et al. (2020). Effect of erythritol on physicochemical properties of reformulated high protein meringues obtained from whey protein isolate.
Materials – please put companies and locations in the brackets according to MDPI standards
please mention all of the materials you use, you have forgotten to provide info on NaOH, phenolophtalein
Line 122 - the mixes were heated for 10 min at 80 °C, then immediately heated cooled down to 4±1 °C
give some space to all the mathematical equations and their descriptions. they are not very readable as they stand.
… … … … … … … … … … … … … why so many dots?
Line 277 - The viscosity level of mixtures is influences influenced
How was the hardness tested?, details please. You mention about the size of the probe, but what about the sample size??
It should be smaller than the probe.
if the probe was smaller than the test sample, then the test was not performed quite correctly. This is not TPA, the description rather suits to back extrusion test. I would change here that you have done a penetration test, where you can also test for these texture parameters, and you will be fine
calculated by the method mentioned in – this phrase is repeated too many times
Please combine RESULTS and DISCUSSION together in your paper, this will make it easier to read and you won't have to repeat the same content
Table 3 - when describing the results, you write about significant differences - except for the glucose value the differences are statistically insignificant. similar is the case in table 4 and table 5
SG – specific gravity, not gravity of specific
Not 0 day – day 1
Lines 374-390 – this is not a discussion, sentences are out of place and should be moved to different sections
Line 416 – Since lactose isn't completely digested in the stomach system, it has a moderately low-energy level – I disagree - lactose has the same calorific value as sucrose
Line 419 - Replacement of sucrose with stevia sucralose, sorbitol and bulking agents had not influences influenced - such errors are repeated many times, too many times
Conclusions
Lines 511-516 – these are not conclusions - just a repetition of the content from the introduction, methodology and summary. Please articulate and highlight the most important observations.
Line - SEM micrographs in frozen yogurt with stevia, sucralose, and sorbitol were exhibited a various microstructure. – this is the most generic sentence as it can get.
Good luck with the corrections:)
Author Response
Top of Form
Open Review
(x) I would not like to sign my review report
( ) I would like to sign my review report
English language and style
(x) Extensive editing of English language and style required
( ) Moderate English changes required
( ) English language and style are fine/minor spell check required
( ) I don't feel qualified to judge about the English language and style
|
Yes |
Can be improved |
Must be improved |
Not applicable |
|
|
Does the introduction provide sufficient background and include all relevant references? |
( ) |
( ) |
(x) |
( ) |
|
Is the research design appropriate? |
( ) |
( ) |
(x) |
( ) |
|
Are the methods adequately described? |
( ) |
(x) |
( ) |
( ) |
|
Are the results clearly presented? |
( ) |
(x) |
( ) |
( ) |
|
Are the conclusions supported by the results? |
( ) |
( ) |
(x) |
( ) |
Comments and Suggestions for Authors
Dear Authors,
The paper penned by Atallah et al. has scientific potential, but in the current form it can not be accepted. The text has to undergo rigorous English editing by native speaker in terms of grammar, vocabulary and punctuation as it is not a reader friendly. There are a lot of errors in the text, there is no point in listing them all, I will just to name a few. There are many commas in the text that are used unjustifiably. Also, please don't use this many semicolons, combine sentences with each other more neatly. I recommend major revision.
Some other comments to improve the manuscript:
Title - you may cross out sensory as these properties fall into physicochemical category
Please do not repeat the same sentences from the summary in the text, some of the same content is duplicated several times.
introduction
Lines 42-46 - Probiotic frozen yogurt products are considered to be a frozen dairy product that includes healthy and nutritious characteristics of ice cream mixed with probiotic-yogurt. Frozen yogurt is a rich source of protein, sugar, and fat levels. The demand of frozen desserts products has currently increased [2,3] especially frozen desserts products without sucrose are increasing [4] .
– please reformulate these sentences, they are very word heavy
Generally, there are many such sentences throughout the text, and in these context the article as a whole needs comprehensive improvement. in addition, the introduction is written chaotically, often there is no logical sequence between the sentences
[4] - reference no 4, concerns vegan ice cream, what does vegan ice cream have to do with the content quoted above? Please cite articles that really do, for example:
Di Monaco, R., Miele, N. A., Cabisidan, E. K., & Cavella, S. (2018). Strategies to reduce
sugars in food. Current Opinion in Food Science, 19, 92-97.
Nastaj et al. (2020). Effect of erythritol on physicochemical properties of reformulated high protein meringues obtained from whey protein isolate.
Materials – please put companies and locations in the brackets according to MDPI standards
please mention all of the materials you use, you have forgotten to provide info on NaOH, phenolophtalein
Line 122 - the mixes were heated for 10 min at 80 °C, then immediately heated cooled down to 4±1 °C
give some space to all the mathematical equations and their descriptions. they are not very readable as they stand.
… … … … … … … … … … … … … why so many dots?
Line 277 - The viscosity level of mixtures is influences influenced
How was the hardness tested?, details please. You mention about the size of the probe, but what about the sample size??
It should be smaller than the probe.
if the probe was smaller than the test sample, then the test was not performed quite correctly. This is not TPA, the description rather suits to back extrusion test. I would change here that you have done a penetration test, where you can also test for these texture parameters, and you will be fine
calculated by the method mentioned in – this phrase is repeated too many times
Please combine RESULTS and DISCUSSION together in your paper, this will make it easier to read and you won't have to repeat the same content
Table 3 - when describing the results, you write about significant differences - except for the glucose value the differences are statistically insignificant. similar is the case in table 4 and table 5
SG – specific gravity, not gravity of specific
Not 0 day – day 1
Lines 374-390 – this is not a discussion, sentences are out of place and should be moved to different sections
Line 416 – Since lactose isn't completely digested in the stomach system, it has a moderately low-energy level – I disagree - lactose has the same calorific value as sucrose
Line 419 - Replacement of sucrose with stevia sucralose, sorbitol and bulking agents had not influences influenced - such errors are repeated many times, too many times
Conclusions
Lines 511-516 – these are not conclusions - just a repetition of the content from the introduction, methodology and summary. Please articulate and highlight the most important observations.
Line - SEM micrographs in frozen yogurt with stevia, sucralose, and sorbitol were exhibited a various microstructure. – this is the most generic sentence as it can get.
Good luck with the corrections:)
Submission Date
01 March 2022
Date of this review
23 Mar 2022 18:47:28
Corrections by author
1- Title - you may cross out sensory as these properties fall into physicochemical category
Au: yes, changed as requested.
- The author changed the title.
- Microstructural, physicochemical and microbiological characteristics of sugar fat free ice cream from buffalo milk
2- Please do not repeat the same sentences from the summary in the text, some of the same content is duplicated several times.
Au: yes, corrected as requested.
- The author reviewed the sentences from the summary in the text and deleted the repeated sentences in the text (abstract, results, discussion, and conclusions) and not repeat the same sentences in the text.
2- Introduction
Lines 42-46 - Probiotic frozen yogurt products are considered to be a frozen dairy product that includes healthy and nutritious characteristics of ice cream mixed with probiotic-yogurt. Frozen yogurt is a rich source of protein, sugar, and fat levels. The demand of frozen desserts products has currently increased [2,3] especially frozen desserts products without sucrose are increasing [4]. – please reformulate these sentences, they are very word heavy
Au: yes, changed and corrected as requested.
- The author changed the sentences in the text as follows.
- Probiotic frozen yogurts are a frozen dairy product and that contain many healthy and nutritious characteristics. Frozen yogurt is a rich source of protein, sugar, and fat levels. The demand of frozen desserts products has currently increased [2,3] especially frozen desserts products without sucrose [4].
3- Generally, there are many such sentences throughout the text, and in these context the article as a whole needs comprehensive improvement. in addition, the introduction is written chaotically, often there is no logical sequence between the sentences
[4] - reference no 4, concerns vegan ice cream, what does vegan ice cream have to do with the content quoted above? Please cite articles that really do, for example:
Di Monaco, R., Miele, N. A., Cabisidan, E. K., & Cavella, S. (2018). Strategies to reduce
sugars in food. Current Opinion in Food Science, 19, 92-97.
Nastaj et al. (2020). Effect of erythritol on physicochemical properties of reformulated high protein meringues obtained from whey protein isolate.
Au: yes, changed and corrected as requested.
- The author reviewed and deleted the sentence [4] - reference no 4, concerns vegan ice cream.
- The author used the cite articles that really do, for example:
Di Monaco, R., Miele, N. A., Cabisidan, E. K., & Cavella, S. (2018). Strategies to reduce sugars in food. Current Opinion in Food Science, 19, 92-97.
4- Materials – please put companies and locations in the brackets according to MDPI standards
Au: yes, changed and corrected as requested.
- The author put companies and locations in the brackets according to MDPI standards in the materials of the text.
5- please mention all of the materials you use, you have forgotten to provide info on NaOH, phenolophtalein
Au: yes, changed and corrected as requested.
- The author reviewed and added all materials especially the NaOH, phenolophtalein in the materials of the text as follows:
Sodium hydroxide (NaOH) and Phenolphthalein were obtained from (Oxoid, Basingstoke, UK).
6- Line 122 - the mixes were heated for 10 min at 80 °C, then immediately heated cooled down to 4±1 °C
Au: yes, changed and corrected as requested.
- The author reviewed and changed the sentence as follows:
the mixes were heated for 10 min at 80 °C, then immediately cooled down to 4±1 °C and aged at 4±1 °C for 24 h.
7- give some space to all the mathematical equations and their descriptions. they are not very readable as they stand.
Au: yes, changed and corrected as requested.
- The author reviewed and given some space to the mathematical equations and their descriptions in all the text.
8- … … … … … … … … … … … … … why so many dots?
Au: yes, changed and corrected as requested.
- The author deleted some dots in all the mathematical equations of the text.
9- Line 277 - The viscosity level of mixtures is influences influenced
Au: yes, changed and corrected as requested.
- The author reviewed and changed the sentence in the text as follows:
The viscosity level of frozen yogurt mixtures before whipping and freezing is influenced with some factors including the stabilizer, protein, fat, bulking agent, and the materials quality used.
10- How was the hardness tested?, details please. You mention about the size of the probe, but what about the sample size??
It should be smaller than the probe.
if the probe was smaller than the test sample, then the test was not performed quite correctly. This is not TPA, the description rather suits to back extrusion test. I would change here that you have done a penetration test, where you can also test for these texture parameters, and you will be fine
Au: yes, changed and corrected as requested.
- The author reviewed the hardness method as follows:
The hardness of frozen yogurt after whipping and freezing was measured using a Universal Testing Machine (TMS-Pro, Tokyo, Japan) equipped with (250 lbf) load cell and linked to a computer programmed with Texture ProTM (Texture ProTM, program, DEVTPA with the hold). A flat rod probe (49.95 mm) was used to uniaxial compress the samples (50 g) to 50% of their original height. The hardness was adjusted to a test speed 60 mm s-1; trigger force 1 N, deformation 25%, temperature at -20 °C, and holding time 2 s between cycles. The hardness value by Universal Testing Machine was expressed as Newton (N).
11- calculated by the method mentioned in – this phrase is repeated too many times
Au: yes, changed and corrected as requested.
- The author reviewed and rewritten this phrase in the text as follows:
as described by
or by the method of
or using
or according to
or by the method in the
12- Please combine RESULTS and DISCUSSION together in your paper, this will make it easier to read and you won't have to repeat the same content
Au:
- In the brackets according to MDPI standards separated RESULTS about DISCUSSION.
The author written this paper according to MDPI standard. RESULTS are section and DISCUSSION is other section.
13- Table 3 - when describing the results, you write about significant differences - except for the glucose value the differences are statistically insignificant. similar is the case in table 4 and table 5
Au: yes, changed and corrected as requested.
- The author corrected and rewritten about only significant differences as Table 3, 4 and 5.
14- SG – specific gravity, not gravity of specific
Au: yes, changed and corrected as requested.
- The author corrected this in the text as follows:
specific gravity (SG)
15- Not 0 day – day 1
Au: yes, changed and corrected as requested.
- The author corrected this in the text as follows:
day 1
16- Lines 374-390 – this is not a discussion, sentences are out of place and should be moved to different sections
Au: yes, changed and corrected as requested.
- The author reviewed and moved the sentences to different sections.
17- Line 416 – Since lactose isn't completely digested in the stomach system, it has a moderately low-energy level – I disagree - lactose has the same calorific value as sucrose
Au: yes, changed and corrected as requested.
- The author deleted the sentences from the text.
18- Line 419 - Replacement of sucrose with stevia sucralose, sorbitol and bulking agents had not influences influenced - such errors are repeated many times, too many times
Au: yes, changed and corrected as requested.
- The author corrected the sentences in the text as follows:
Replacement of sucrose with stevia sucralose, sorbitol and bulking agents had not influenced
19- Conclusions
Lines 511-516 – these are not conclusions - just a repetition of the content from the introduction, methodology and summary. Please articulate and highlight the most important observations.
Au: yes, changed and corrected as requested.
- The author reviewed and rewritten the conclusions as follows:
In this study, four different probiotic-frozen yogurts were produced (F1, F2, F3 and F4). The characteristics of physical, textural, and sensory of frozen yogurt are important quality characteristics because they play an important role in consumers’ acceptance of these products. Chemical characteristics were not significantly affected by replacement of sucrose with sweeteners and bulking agents compared with the control samples. Viscosity, overrun, and hardness values were significantly affected by sweeteners in the treatments. The Str. thermophilus and Lb. bulgaricus counts were decreased gradually with prolong storage periods (p < 0.05). In all treatments, Bifidobacterium bifidium counts varied from 7.56 to 7.60 log10 CFU g-1 at 1-day and these counts were lowered gradually with prolong storage periods (p < 0.05), but the counts remained viable (> 6.00 log10 CFU g-1) up to 60 d. In the F1, F2 and F3 treatments, the highest values of total acceptability were detected during storage periods. SEM micrographs in all frozen yogurt treatments were exhibited a various microstructure. The microstructure characteristics were improved in probiotic-frozen yogurts with stevia, sucralose and sorbitol compared with the control. However, further study is necessary to further replace the fat and sugar to produce the low-calorie frozen yogurt and determining the keeping quality of these products. Now, Egyptian customers are more aware of their health status, and hence conscious of free-fat and -sugar frozen yogurt diets.
20- Line - SEM micrographs in probiotic-frozen yogurts with stevia, sucralose, and sorbitol were exhibited a various microstructure. – this is the most generic sentence as it can get.
Au: yes, changed and corrected as requested.
- The author reviewed and rewritten the sentence as follows:
SEM micrographs in all frozen yogurt treatments were exhibited a various microstructure. The microstructure characteristics were improved in probiotic-frozen yogurts with stevia, sucralose and sorbitol compared with the control.

Reviewer 2 Report
It's interesting how the authors decided to pick buffalo milk, is there a reason why?
A process diagram would benefit the readers to summarise Section 2.3
How did the authors decide the level of different sweeteners in their samples?
Is it possible for the authors to calculate cost to produce per 100g to provide some context as well for industry manufacturers?
Formula 4 can be simplified by placing the words in the equation?
Section 2.7 - the authors mention hedonic so it this liking score? How can the panellists rate icy structure this way?
Samples were provided @20C? I thought this was frozen yoghurt?
How did the authors train their panel?
Section 2.9
What's qPCR datasets?
What posthoc did the authors attempted?
Figure 2 - since the line scale goes to 10, please adjust the 10.2. Also provide a break on the graph like 0 // as the scales doesnt start at 8
Figure 3 should go before Discussion
Discussion is fine where the authors have attempted to explain their results.
What were the limitations and future research avenue? This should be added in the Conclusion.
Author Response
Top of Form
Open Review
(x) I would not like to sign my review report
( ) I would like to sign my review report
English language and style
( ) Extensive editing of English language and style required
( ) Moderate English changes required
(x) English language and style are fine/minor spell check required
( ) I don't feel qualified to judge about the English language and style
|
Yes |
Can be improved |
Must be improved |
Not applicable |
|
|
Does the introduction provide sufficient background and include all relevant references? |
( ) |
(x) |
( ) |
( ) |
|
Is the research design appropriate? |
( ) |
( ) |
(x) |
( ) |
|
Are the methods adequately described? |
( ) |
( ) |
(x) |
( ) |
|
Are the results clearly presented? |
( ) |
(x) |
( ) |
( ) |
|
Are the conclusions supported by the results? |
( ) |
(x) |
( ) |
( ) |
Comments and Suggestions for Authors
It's interesting how the authors decided to pick buffalo milk, is there a reason why?
A process diagram would benefit the readers to summarise Section 2.3
How did the authors decide the level of different sweeteners in their samples?
Is it possible for the authors to calculate cost to produce per 100g to provide some context as well for industry manufacturers?
Formula 4 can be simplified by placing the words in the equation?
Section 2.7 - the authors mention hedonic so it this liking score? How can the panellists rate icy structure this way?
Samples were provided @20C? I thought this was frozen yoghurt?
How did the authors train their panel?
Section 2.9
What's qPCR datasets?
What posthoc did the authors attempted?
Figure 2 - since the line scale goes to 10, please adjust the 10.2. Also provide a break on the graph like 0 // as the scales doesnt start at 8
Figure 3 should go before Discussion
Discussion is fine where the authors have attempted to explain their results.
What were the limitations and future research avenue? This should be added in the Conclusion.
Submission Date
01 March 2022
Date of this review
17 Mar 2022 12:15:57
Bottom of Form
© 1996-2022 MDPI (Corrections by author
1- It's interesting how the authors decided to pick buffalo milk, is there a reason why?
Au:
- There is not a reason to pick buffalo milk in this study. The herd of Agriculture Faculty (Moshtohor, Benha University, Qalubia, Egypt) contain buffalo milk and authors used this milk in the manufacture.
2- A process diagram would benefit the readers to summarise Section 2.3
Au: Corrected as requested.
- The author changed the Section 2.3 to a process diagram as requested.
3- How did the authors decide the level of different sweeteners in their samples?
Au: yes, written as requested
- The author used initial trials and written in the section 2.3 as following:
Initial trials were found in the produced frozen yogurt with different levels of stevia, sorbitol, and sucralose. Results were detected that the best value of Stevia was 0.06%, sucralose was 0.03%, and sorbitol was 3% in the produced frozen yogurt with determining some analyses such as sensory and texture analyses. Therefore, initial trials were found in the produced frozen yogurt with different levels of maltodextrin and polydextrose as filler. Data was found that the best concentration of maltodextrin was 7.5%, and polydextrose was 7.5% in the frozen yogurt manufacture (results not shown).
4- Is it possible for the authors to calculate cost to produce per 100g to provide some context as well for industry manufacturers?
Au: yes, calculated as requested
- The author calculated cost of frozen yogurts (per 100g) according to reviewer as follows:
|
Ingredients |
Control frozen yogurt |
Frozen yogurt with stevia |
Frozen yogurt with sucralose |
Frozen yogurt with sorbitol |
|
Sugar |
150 |
0 |
0 |
0 |
|
Stabilizer (CMC) |
2.5 |
2.5 |
2.5 |
2.5 |
|
Full powder milk |
56.5 |
56.5 |
56.5 |
56.5 |
|
Buffaloes milk |
738 |
738 |
738 |
738 |
|
Fresh cream |
53 |
53 |
53 |
53 |
|
Polydextrose |
0 |
74.7 |
74.85 |
60 |
|
Maltodetrin |
0 |
74.7 |
74.85 |
60 |
|
Stevia |
0 |
0.6 |
0 |
0 |
|
Sucralose |
0 |
0 |
0.3 |
0 |
|
Sorbitol |
0 |
0 |
0 |
30 |
|
Total |
1000 |
1000 |
1000 |
1000 |
|
Total cost/100g (Egyptian pound) |
4.9 |
5 |
5.2 |
5 |
5- Formula 4 can be simplified by placing the words in the equation?
Au:
- If the author placed the words in the equation will take more lines and this will give the equation of shape unacceptable.
- The author simplified by placing the words in the equation as follows:
where: A is weight of mix volume before whipping and freezing and B: weight of the same volume of frozen yogurt sample after whipping and freezing.
6- Section 2.7 - the authors mention hedonic so it this liking score? How can the panellists rate icy structure this way?
Au: yes, written as requested
-The panels were evaluated the icy structure by the mouse and eyes.
7- Samples were provided @20C? I thought this was frozen yoghurt?
Au: yes, corrected as requested.
- The probiotic-frozen yogurt samples were coded and given to each panelist at -20 °C.
8- How did the authors train their panel?
Au: yes, written as requested
- Panels (15) were selected based on their interest in the evaluation of frozen yogurt and were trained by testing commercial frozen yogurt. Samples (50 g) were submitted in a group to the 15 test panelists. The panels were trained many times on the evaluation of frozen yogurt.
9- Section 2.9
What's qPCR datasets?
Au:
-The qPCR datasets are Quantitative PCR datasets and the author written in the section 2.9.
10- Figure 2 - since the line scale goes to 10, please adjust the 10.2. Also provide a break on the graph like 0 // as the scales doesnt start at 8
Au: yes, corrected as requested.
The author adjusted the format axis of Figure 2 according to the reviewer.
11- Figure 3 should go before Discussion
Au: yes, corrected as requested.
- Figure 3 is placed before Discussion
12- Discussion is fine where the authors have attempted to explain their results.
Au:
No comment of reviewer.
13- What were the limitations and future research avenue? This should be added in the Conclusion.
Au: yes, written as requested.
- The author added the limitations and future research avenue in the conclusion as follows:
However, further study is necessary to further replace the fat and sugar to produce the low-calorie frozen yogurt and determining the keeping quality of these products. Now, Egyptian customers are more aware of their health status, and hence conscious of free-fat and -sugar frozen yogurt diets.

Round 2
Reviewer 1 Report
Dear Authors,
The authors have implemented most of the substantive revisions I suggested. I think that in this form the article can be recommended for publication. However, the text still requires solid revisions in terms of English. I have pointed out this before. I ask provocatively: has the article been proofread by native speaker? Because I know it has not been done.
So let the final decision about acceptance be made by editor-in-chief.
Good luck!
Author Response
Top of Form
Top of Form
Open Review
(x) I would not like to sign my review report
( ) I would like to sign my review report
English language and style
(x) Extensive editing of English language and style required
( ) Moderate English changes required
( ) English language and style are fine/minor spell check required
( ) I don't feel qualified to judge about the English language and style
|
Yes |
Can be improved |
Must be improved |
Not applicable |
|
|
Does the introduction provide sufficient background and include all relevant references? |
(x) |
( ) |
( ) |
( ) |
|
Is the research design appropriate? |
(x) |
( ) |
( ) |
( ) |
|
Are the methods adequately described? |
(x) |
( ) |
( ) |
( ) |
|
Are the results clearly presented? |
(x) |
( ) |
( ) |
( ) |
|
Are the conclusions supported by the results? |
( ) |
(x) |
( ) |
( ) |
Comments and Suggestions for Authors
Dear Authors,
The authors have implemented most of the substantive revisions I suggested. I think that in this form the article can be recommended for publication. However, the text still requires solid revisions in terms of English. I have pointed out this before. I ask provocatively: has the article been proofread by native speaker? Because I know it has not been done.
Good luck!
Submission Date
01 March 2022
Date of this review
30 Mar 2022 21:23:24Bottom of Form
Corrections by author
Comments and Suggestions for Authors
Dear Authors,
The authors have implemented most of the substantive revisions I suggested. I think that in this form the article can be recommended for publication. However, the text still requires solid revisions in terms of English. I have pointed out this before. I ask provocatively: has the article been proofread by native speaker? Because I know it has not been done.
Au: yes, the author reviewed and changed as local speaker.
The manuscript was given to a local speaker of the English language, and he reviewed the manuscript according to the requirements of the reviewers, and all the notes of the local speaker on the manuscript were recorded in the first revision, as the reviewer said.

Reviewer 2 Report
Bufallo milk isn't a common milk consumed in the Western country, but rather seems to be very common in Egypt where the authors are located. Perhaps a couple of statements on familiarity would be beneficial to the readers.
More details is still required, what did the authors exactly do with the panel to train them for the commercial frozen yoghurt? Was there a calibration? Or what was done?
Other than that, the manuscript is suitable for publication.
Author Response
Top of Form
Open Review
(x) I would not like to sign my review report
( ) I would like to sign my review report
English language and style
( ) Extensive editing of English language and style required
( ) Moderate English changes required
(x) English language and style are fine/minor spell check required
( ) I don't feel qualified to judge about the English language and style
|
Yes |
Can be improved |
Must be improved |
Not applicable |
|
|
Does the introduction provide sufficient background and include all relevant references? |
( ) |
(x) |
( ) |
( ) |
|
Is the research design appropriate? |
( ) |
(x) |
( ) |
( ) |
|
Are the methods adequately described? |
( ) |
( ) |
(x) |
( ) |
|
Are the results clearly presented? |
( ) |
(x) |
( ) |
( ) |
|
Are the conclusions supported by the results? |
( ) |
(x) |
( ) |
( ) |
Comments and Suggestions for Authors
Bufallo milk isn't a common milk consumed in the Western country, but rather seems to be very common in Egypt where the authors are located. Perhaps a couple of statements on familiarity would be beneficial to the readers.
More details is still required, what did the authors exactly do with the panel to train them for the commercial frozen yoghurt? Was there a calibration? Or what was done?
Other than that, the manuscript is suitable for publication.
Submission Date
01 March 2022
Date of this review
28 Mar 2022 18:01:02
Bottom of Form
© 1996-2022 MDPI (Corrections by author
Comments and Suggestions for Authors
1- Bufallo milk isn't a common milk consumed in the Western country, but rather seems to be very common in Egypt where the authors are located. Perhaps a couple of statements on familiarity would be beneficial to the readers.
Au: yes, noticed as requested.
Buffalo milk has many benefits:
- The increase in the percentage of total solids reaches from 16-18%.
Fat content increases from 5-8%.
Protein content increased from 3-4%.
Its high content of solids compared to cow's milk such as vitamins - salts and lactose sugar.
2- More details is still required, what did the authors exactly do with the panel to train them for the commercial frozen yoghurt? Was there a calibration? Or what was done?
Au: yes, noticed as requested.
As for the training of arbitrators on evaluation, it is carried out through the following:
1- All arbitrators must be free of diseases and be periodically examined.
2- Training courses are held periodically for them.
3- The ages of the arbitrators are chosen between 20 – 55 years
4- Practical training is conducted for them as follows:
A- Introduce them to frozen yogurt
B- Features of frozen yogurt
C- The most important characteristics of frozen yogurt
D- Frozen Yogurt Judging Forms
E- Offer them commercial frozen yogurt and they judge
F- Conducting a review of specialists on arbitration
G- Making a feedback for the judges on frozen yogurt and clarifying the most important points
H- Re-arbitration and evaluation of the same products again
I- Re-evaluation of the arbitrators
J- The panelist given certificates and approval to the arbitrators who passed the arbitration.

This manuscript is a resubmission of an earlier submission. The following is a list of the peer review reports and author responses from that submission.
Round 1
Reviewer 1 Report
1-What is criteria for selection of stevia, sucralose, and sorbitol?
Similarity 38% is not acceptable. Regarding the attached file, many sentences have been copied and paste.
2-Line 23, P<0.05 is not correct. How about oint p=0.05?
3-What was the practical conclusion from the following sentence?
Scanning electron microscopy (SEM) micrographs of frozen yogurt containing stevia, sucralose and sorbitol, were illustrated that the frozen yogurt gel exhibited a various structure with a fine network and containednumbers of very small pores in size.
So what?
4-Why total count of just these strains (Str. thermophilus and Lb. delbrueckii ssp. Bulgaricus & bifido) has been checked? Are they the best selected probiotics for addition to ice cream and frozen products? Was not better to check viability of other psychrophilic probiotics?
5- Some statements need references.
6- Cross references is a big mistake. Please avoid using a second reference instead of citation to main one. Just for instance: Frozen yogurt is known as ice cream with some properties of yogurt [1].
While the reference number 1 is a second reference which has cited to the main one. "Soukoulis, C., Tzia, C. Response surface mapping of the sensory characteristics and acceptability of chocolate ice cream con-567 taining alternate sweetening agents. J. Sensory Studies, 2010, 25, 50-75. "
7- There are repeated words in some sentences e.g. Line 18-19 (containing…) or in Conclusion 538-538
8-Gap of research should be described.
9- Line 526: The text is a mixture of two reference style: e.g. line 526 "Alizadeh et al. (2014) [47]……".
10- I think one important parameters on products characteristics is formulation especially fat. Why the impact of process variables are not investigated?
11- There are many similar paper not cited in the text.

Author Response
Corrections of comments
1-What is criteria for selection of stevia, sucralose, and sorbitol?
Au:
- These are some criteria for selection of stevia, sucralose, and sorbitol as follows:
1- stevia, sucralose, and sorbitol are recognized as safe in the reports of JECFA, WHO and FDA.
2- stevia and sorbitol are natural materials.
3- sucralose is an artificial sweetener and approved safe.
4- these sweeteners are a very low-calorie.
5- these sweeteners are sweetened more than sugar.
6- these sweeteners are used in many foods and dairy products.
7- these sweeteners are extended in the world.
Similarity 38% is not acceptable. Regarding the attached file, many sentences have been copied and paste.
Au:
- The authors are very sorry about this.
- The authors are rewriting the manuscript.
2-Line 23, P<0.05 is not correct. How about oint p=0.05?
Au:
- The author changes of the sentences.
- The highest viscosity, overrun and hardness values (p ≤ 0.05) were recorded
3-What was the practical conclusion from the following sentence?
Scanning electron microscopy (SEM) micrographs of frozen yogurt containing stevia, sucralose and sorbitol, were illustrated that the frozen yogurt gel exhibited a various structure with a fine network and contained numbers of very small pores in size.
Au:
- The author changes of the sentences.
- Scanning electron microscopy (SEM) micrographs of frozen yogurt containing stevia, sucralose, and sorbitol, were illustrated that the microstructure of frozen yogurt samples was shown a difference with a fine network, and the images were contained very low pores in size for the F2, F3, and F4 samples.
So what?
- Au:
- These results appeared that the use these sweeteners and bulk agent were improved the microstructure of frozen yogurts.
4-Why total count of just these strains (Str. thermophilus and Lb. delbrueckii ssp.Bulgaricus & bifido) has been checked? Are they the best selected probiotics for addition to ice cream and frozen products? Was not better to check viability of other psychrophilic probiotics?
- Au:
- The frozen yogurt is consisted of yogurt and ice cream.
- yogurt is made from yogurt starter and the yogurt starter is contained two strains including Str. thermophilus and Lb. delbrueckii ssp. bulgaricus also, the best selected probiotics for addition to frozen yogurt is Bifidobacterium bifidium.
- no other strains are used in this study.
- the microbiology examinations will count the addition strains in this study and the strains are Str. Thermophilus, Lb. delbrueckii ssp. bulgaricus and Bifidobacterium bifidium.
- other psychrophilic probiotics is not found in this study.
5- Some statements need references.
- Au:
- The author adds some references in some statements for the study.
6- Cross references is a big mistake. Please avoid using a second reference instead of citation to main one. Just for instance: Frozen yogurt is known as ice cream with some properties of yogurt [1].
- Au:
- The author reviewed of the manuscript.
- The author corrected these comments in the manuscript.
While the reference number 1 is a second reference which has cited to the main one. "Soukoulis, C., Tzia, C. Response surface mapping of the sensory characteristics and acceptability of chocolate ice cream con-567 taining alternate sweetening agents. J. Sensory Studies, 2010, 25, 50-75. "
- Au:
- The author reviewed of the manuscript.
- The author corrected these comments in the manuscript.
7- There are repeated words in some sentences e.g. Line 18-19 (containing…) or in Conclusion 538-538
- Au:
- The author reviewed of the manuscript.
- The author corrected these comments in the manuscript.
8-Gap of research should be described.
- Au:
-- The author noticed in the abstract of the manuscript.
Frozen yogurt is known as ice cream with some properties of yogurt. It is a rich source of sugar levels between 15% and 28% of the total materials. Consumers suffering from lactose intolerance, and metabolic syndrome, are looking for low sugar and low calories products.
- The author noticed in the introduction of the manuscript.
Frozen yogurt is a rich source of protein, sugar, and fat levels. The demand of frozen desserts products has currently increased [2,3] especially frozen desserts products without sucrose are increasing [4]. Sucrose level in frozen dessert products was changed between 9% and 28% of the total materials [5]. Consumers suffering from lactose intolerance, metabolic syndrome, diabetes, allergies are looking for low sugar and low calories products.
- The author noticed in the aim of the manuscript.
The current study aimed to produce a novel probiotic-frozen yogurt using different sweeteners as a substitution for sugar, and to compare their possibility effects on physicochemical, microbiological, microstructure and sensory characteristics during storage.
9- Line 526: The text is a mixture of two reference style: e.g. line 526 "Alizadeh et al. (2014) [47]……".
- Au:
- The author reviewed of the manuscript.
- The author corrected these comments in the manuscript.
10- I think one important parameters on products characteristics is formulation especially fat. Why the impact of process variables are not investigated?
- Au:
- The manuscript was studied replacement of sugar with some sweeteners, but the fat content is not changed in the study. The important parameters in this study related to replacement of sugar.
Why the impact of process variables are not investigated?
- Au:
- The manuscript was not studied replacement of fat in this study, but the fat content is not changed in all treatments (F1 as a control frozen yogurt, F2 as frozen yogurt with stevia, F3 as frozen yogurt with sucralose, and F4 as frozen yogurt with sorbitol), followed the variables are not changed.
11- There are many similar paper not cited in the text.
- Au:
- The author noticed the important similar paper in this study.

Reviewer 2 Report
Dear Authors
I think the paper in the current form lacks from novelty existence. And the color and viscosity must be cosnidered.
Design of experiment is not clear. The effect of variables are presumbly various beforehand the implementation. The paper must focus on viscosity and color parameters. Also the used instrumental for textural parameters was old fashioned.
Author Response
Open Review
( ) I would not like to sign my review report
(x) I would like to sign my review report
English language and style
( ) Extensive editing of English language and style required
(x) Moderate English changes required
( ) English language and style are fine/minor spell check required
( ) I don't feel qualified to judge about the English language and style
|
Yes |
Can be improved |
Must be improved |
Not applicable |
|
|
Does the introduction provide sufficient background and include all relevant references? |
( ) |
(x) |
( ) |
( ) |
|
Is the research design appropriate? |
(x) |
( ) |
( ) |
( ) |
|
Are the methods adequately described? |
(x) |
( ) |
( ) |
( ) |
|
Are the results clearly presented? |
( ) |
(x) |
( ) |
( ) |
|
Are the conclusions supported by the results? |
( ) |
( ) |
(x) |
( ) |
Comments and Suggestions for Authors
Dear Authors
I think the paper in the current form lacks from novelty existence. And the color and viscosity must be cosnidered.
Design of experiment is not clear. The effect of variables are presumbly various beforehand the implementation. The paper must focus on viscosity and color parameters. Also the used instrumental for textural parameters was old fashioned.
Submission Date
20 January 2022
Date of this review
04 Feb 2022 19:51:15
Corrections of comments
1- I think the paper in the current form lacks from novelty existence.
Au:
- The author reviewed of the manuscript.
- The research does not lose its seriousness, but on the contrary, the research studies one of the most important problems in the frozen yogurt industry, which is the high percentage of added sugar in the industry, which negatively affects the health of the consumer of all categories and affiliations, especially diabetes, obesity, and allergies. All of this was mentioned in the research paper and the research was designed to find Solutions to this problem through the complete replacement of sugar with some sweeteners as substitutes for sugar, and many characteristics and tests were used to measure the extent of the positive or negative effect. The research also contains a review of previous research in this field.
2- And the color and viscosity must be cosnidered.
Au:
- The author reviewed of the manuscript.
- The research dealt with the study of viscosity in detail by mentioning the method used in the measurement in the section on materials and methods of frozen yogurt. In addition to a detailed explanation of it in the results and discussions, please refer to the physical properties in the methods, results, and discussions.
- The paper dealt with the color and appearance of the different transactions in the research, in the part of sensory characteristics in detail, and this was shown in the results and discussion. Please review the sensory properties section of methods, results, and discussions.
3- Design of experiment is not clear.
Au:
- The author reviewed of the manuscript.
- The design of the paper is as follows:
The research consists of the title, summary, introduction, problem and objective of the research, materials, methods, results, discussions, and finally the research summary.
If each of the parts is discussed in detail, we find the following:
First, the title is quite clear (Please refer to the title).
Second: The summary reviews an overview of frozen yogurt and the different treatments for frozen yogurt and reviews the different measurements and characteristics. Then the most important results of the research were reviewed (please refer to the introduction part).
(Please refer to the abstract part).
Third: An overview of the most important previous studies in the field of research was reviewed (please refer to the introduction part).
Fourth: The most important problems and the aim of the research were reviewed. (Probiotic-frozen yogurt is frozen dairy products which includes healthy and nutritious aspects. Probiotic-frozen yogurts are a rich source of sugar, and fat. The sucrose limit in probiotic-frozen yogurt changes between 15% and 28% of the total components. Due to the high prevalence of Type 2 diabetes and obesity among children and adolescents, people are now more aware of their health status, and hence conscious of their diet. This health-conscious decision poses a formidable challenge to probiotic-frozen yogurt production. As a result, the probiotic-frozen yogurt market trend is moving towards a sugar-free probiotic-frozen yogurt formulation with excellent texture, structure, and sensory attributes to gain consumer satisfaction. The current study aimed to produce a novel probiotic-frozen yogurt using different sweeteners as a substitution for sugar, and to compare their possibility effects on physicochemical, microbiological, microstructure and sensory characteristics during storage).
Fifth: Materials and methods: A detailed review of all materials and methods used in the field of study (please refer to the materials and methods part).
Sixth: Results: The results were reviewed in detail for all the results obtained from the research paper (please refer to the results part).
Seventh: Discussions: Detailed discussions of the results obtained by comparing them with the results of other scientists' research were reviewed (please refer to the discussions part).
Eighth: The research summary that presents the most important results and the final recommendation of the research (please refer to the conclusions part).
4- The effect of variables are presumbly various beforehand the implementation.
Au:
- The author reviewed of the effect of variables are presumably various beforehand the implementation in the manuscript.
-The study illustrates this by mentioning the variables under study, represented by the different characteristics that were measured (chemical, physical, microbiological, microstructure and sensory). In addition to this, the various factors under study are the factor of different transactions and the factor of storage.
5- The paper must focus on viscosity and color parameters.
- Au:
- The author reviewed of the manuscript.
- The research dealt with the study of viscosity in detail by mentioning the method used in the measurement in the section on materials and methods. In addition to a detailed explanation of it in the results and discussions, please refer to the physical properties in the methods, results, and discussions (please refer to the materials, methods, results, and discussions parts).
- The paper dealt with the color and appearance of the different transactions in the research, in the part of sensory characteristics in detail, and this was shown in the results and discussion. Please review the sensory properties section of methods, results, and discussions. (Please refer to the materials, methods, results, and discussions parts).
6- Also the used instrumental for textural parameters was old fashioned.
Au:
- The author reviewed of the manuscript.
-The estimation of the texture was done using the latest methods as well as the use of the latest device, where only one characteristic, which is the hardness, was estimated.
But there is another research team working on estimating the rest of the properties such as flexibility, tenderness, juiciness, roughness, chewiness, and other properties. In addition to appreciating some other attributes.

Reviewer 3 Report
It would be reasonable to consider adjusting the topic of the work in such a way as not to duplicate the concepts ... it is obvious to me that if he adds sweeteners it is in order to eliminate sucrose.
line 65-77 it would be good to mention ADI for each of the sweeteners
the bibliography requires correction, duplicate numbers in some of the quoted entries
Developed new products, according to the introduction, are dedicated to people who want to reduce the amount of calories consumed - by reducing or eliminating sucrose. Therefore, it would be appropriate to provide an estimate of the caloric value of the kcal. Often many consumers fall into the trap of light products - they think that they can eat much more than those containing, for example, sucrose. I think that CMC and other substances stabilizing the structure of the product can unfortunately increase or equal the caloric content.
I would also ask you to specify to what extent the people selected for the sensory panel were "related" to the product line 215 "... were selected based on their interest in the sensory evaluations of frozen gurt and were trained by testing commercial frozen yogurt". It is possible that a mistake was made in selecting people for the sensory panel.
Author Response
Open Review
(x) I would not like to sign my review report
( ) I would like to sign my review report
English language and style
( ) Extensive editing of English language and style required
( ) Moderate English changes required
( ) English language and style are fine/minor spell check required
(x) I don't feel qualified to judge about the English language and style
|
Yes |
Can be improved |
Must be improved |
Not applicable |
|
|
Does the introduction provide sufficient background and include all relevant references? |
(x) |
( ) |
( ) |
( ) |
|
Is the research design appropriate? |
(x) |
( ) |
( ) |
( ) |
|
Are the methods adequately described? |
( ) |
(x) |
( ) |
( ) |
|
Are the results clearly presented? |
( ) |
(x) |
( ) |
( ) |
|
Are the conclusions supported by the results? |
(x) |
( ) |
( ) |
( ) |
Comments and Suggestions for Authors
It would be reasonable to consider adjusting the topic of the work in such a way as not to duplicate the concepts ... it is obvious to me that if he adds sweeteners it is in order to eliminate sucrose.
line 65-77 it would be good to mention ADI for each of the sweeteners
the bibliography requires correction, duplicate numbers in some of the quoted entries
Developed new products, according to the introduction, are dedicated to people who want to reduce the amount of calories consumed - by reducing or eliminating sucrose. Therefore, it would be appropriate to provide an estimate of the caloric value of the kcal. Often many consumers fall into the trap of light products - they think that they can eat much more than those containing, for example, sucrose. I think that CMC and other substances stabilizing the structure of the product can unfortunately increase or equal the caloric content.
I would also ask you to specify to what extent the people selected for the sensory panel were "related" to the product line 215 "... were selected based on their interest in the sensory evaluations of frozen gurt and were trained by testing commercial frozen yogurt". It is possible that a mistake was made in selecting people for the sensory panel.
Submission Date
20 January 2022
Date of this review
30 Jan 2022 09:58:27
, Switzerland) unless otherwise stated
Corrections of comments
1- Title
It would be reasonable to consider adjusting the topic of the work in such a way as not to duplicate the concepts ... it is obvious to me that if he adds sweeteners it is in order to eliminate sucrose.
Au:
- The author changes of the title.
- Chemical, Physical, Microbiological, Microstructural, and Sensory characteristics of Sucrose Free Probiotic-Frozen Yogurt
2- line 65-77 it would be good to mention ADI for each of the sweeteners
Au:
- stevia, sucralose, and sorbitol are recognized as safe in the reports of JECFA, WHO and FDA.
3- the bibliography requires correction, duplicate numbers in some of the quoted entries
Au:
- The author reviewed of the manuscript.
- The author corrected these comments in the manuscript.
4- Developed new products, according to the introduction, are dedicated to people who want to reduce the amount of calories consumed - by reducing or eliminating sucrose. Therefore, it would be appropriate to provide an estimate of the caloric value of the kcal. Often many consumers fall into the trap of light products - they think that they can eat much more than those containing, for example, sucrose. I think that CMC and other substances stabilizing the structure of the product can unfortunately increase or equal the caloric content.
Au:
- The author reviewed and corrected of the manuscript.
- CMC and other substances stabilizing is adding very small levels (0.1 -0.5%) in frozen yogurt but the sugar is adding very higher levels (15%).
5- I would also ask you to specify to what extent the people selected for the sensory panel were "related" to the product line 215 "... were selected based on their interest in the sensory evaluations of frozen gurt and were trained by testing commercial frozen yogurt". It is possible that a mistake was made in selecting people for the sensory panel.
Au:
- In the Department of Dairy Science contained section of sensory evaluation. This section contained the sensory panels. These panels selected for the sensory evaluation of dairy products. These panels were trained by testing commercial products before the evaluation. For example, in frozen yogurt, the panels were trained on commercial frozen yogurt before the produced frozen yogurt.
- Also, the panels are the stuffs in Dairy Science Department, and the panels are the high experiences in this field.
Open Review
(x) I would not like to sign my review report
( ) I would like to sign my review report
English language and style
( ) Extensive editing of English language and style required
( ) Moderate English changes required
( ) English language and style are fine/minor spell check required
(x) I don't feel qualified to judge about the English language and style
|
Yes |
Can be improved |
Must be improved |
Not applicable |
|
|
Does the introduction provide sufficient background and include all relevant references? |
(x) |
( ) |
( ) |
( ) |
|
Is the research design appropriate? |
(x) |
( ) |
( ) |
( ) |
|
Are the methods adequately described? |
( ) |
(x) |
( ) |
( ) |
|
Are the results clearly presented? |
( ) |
(x) |
( ) |
( ) |
|
Are the conclusions supported by the results? |
(x) |
( ) |
( ) |
( ) |
Comments and Suggestions for Authors
It would be reasonable to consider adjusting the topic of the work in such a way as not to duplicate the concepts ... it is obvious to me that if he adds sweeteners it is in order to eliminate sucrose.
line 65-77 it would be good to mention ADI for each of the sweeteners
the bibliography requires correction, duplicate numbers in some of the quoted entries
Developed new products, according to the introduction, are dedicated to people who want to reduce the amount of calories consumed - by reducing or eliminating sucrose. Therefore, it would be appropriate to provide an estimate of the caloric value of the kcal. Often many consumers fall into the trap of light products - they think that they can eat much more than those containing, for example, sucrose. I think that CMC and other substances stabilizing the structure of the product can unfortunately increase or equal the caloric content.
I would also ask you to specify to what extent the people selected for the sensory panel were "related" to the product line 215 "... were selected based on their interest in the sensory evaluations of frozen gurt and were trained by testing commercial frozen yogurt". It is possible that a mistake was made in selecting people for the sensory panel.
Submission Date
20 January 2022
Date of this review
30 Jan 2022 09:58:27
, Switzerland) unless otherwise stated
Corrections of comments
1- Title
It would be reasonable to consider adjusting the topic of the work in such a way as not to duplicate the concepts ... it is obvious to me that if he adds sweeteners it is in order to eliminate sucrose.
Au:
- The author changes of the title.
- Chemical, Physical, Microbiological, Microstructural, and Sensory characteristics of Sucrose Free Probiotic-Frozen Yogurt
2- line 65-77 it would be good to mention ADI for each of the sweeteners
Au:
- stevia, sucralose, and sorbitol are recognized as safe in the reports of JECFA, WHO and FDA.
3- the bibliography requires correction, duplicate numbers in some of the quoted entries
Au:
- The author reviewed of the manuscript.
- The author corrected these comments in the manuscript.
4- Developed new products, according to the introduction, are dedicated to people who want to reduce the amount of calories consumed - by reducing or eliminating sucrose. Therefore, it would be appropriate to provide an estimate of the caloric value of the kcal. Often many consumers fall into the trap of light products - they think that they can eat much more than those containing, for example, sucrose. I think that CMC and other substances stabilizing the structure of the product can unfortunately increase or equal the caloric content.
Au:
- The author reviewed and corrected of the manuscript.
- CMC and other substances stabilizing is adding very small levels (0.1 -0.5%) in frozen yogurt but the sugar is adding very higher levels (15%).
5- I would also ask you to specify to what extent the people selected for the sensory panel were "related" to the product line 215 "... were selected based on their interest in the sensory evaluations of frozen gurt and were trained by testing commercial frozen yogurt". It is possible that a mistake was made in selecting people for the sensory panel.
Au:
- In the Department of Dairy Science contained section of sensory evaluation. This section contained the sensory panels. These panels selected for the sensory evaluation of dairy products. These panels were trained by testing commercial products before the evaluation. For example, in frozen yogurt, the panels were trained on commercial frozen yogurt before the produced frozen yogurt.
- Also, the panels are the stuffs in Dairy Science Department, and the panels are the high experiences in this field.

Round 2
Reviewer 1 Report
The text has improved in data presentation. But gap of research and discussion can be improved more.
Check similarity of text by attached file.

Author Response
Open Review
( ) I would not like to sign my review report
(x) I would like to sign my review report
English language and style
( ) Extensive editing of English language and style required
( ) Moderate English changes required
( ) English language and style are fine/minor spell check required
(x) I don't feel qualified to judge about the English language and style
|
Yes |
Can be improved |
Must be improved |
Not applicable |
|
|
Does the introduction provide sufficient background and include all relevant references? |
(x) |
( ) |
( ) |
( ) |
|
Is the research design appropriate? |
(x) |
( ) |
( ) |
( ) |
|
Are the methods adequately described? |
(x) |
( ) |
( ) |
( ) |
|
Are the results clearly presented? |
( ) |
(x) |
( ) |
( ) |
|
Are the conclusions supported by the results? |
(x) |
( ) |
( ) |
( ) |
Comments and Suggestions for Authors
The text has improved in data presentation. But gap of research and discussion can be improved more.
Check similarity of text by attached file.
Submission Date
20 January 2022
Date of this review
13 Feb 2022 06:47:10
Corrections
After greeting
All comments and notes sent by the reviewer to the manuscript have been modified.
